# Rheological Behavior of High-Performance Shotcrete Mixtures Containing Colloidal Silica and Silica Fume Using the Bingham Model

**DOI:** 10.3390/ma15020428

**Published:** 2022-01-06

**Authors:** Kyong Ku Yun, Jong Beom Kim, Chang Seok Song, Mohammad Shakhawat Hossain, Seungyeon Han

**Affiliations:** 1Department of Civil Engineering, Kangwon National University, 1 Gangwondaegil, Chuncheon 24341, Korea; kkyun@kangwon.ac.kr (K.K.Y.); jbkim3790@kangwon.ac.kr (J.B.K.); changsuk22c@kangwon.ac.kr (C.S.S.); 2KIIT (Kangwon Institute of Inclusive Technology), Kangwon National University, 1 Gangwondaegil, Chuncheon 24341, Korea

**Keywords:** wet-mix shotcrete, colloidal silica, silica fume, pumpability, shootablilty, ICAR rheometer, Bingham model

## Abstract

There have been numerous studies on shotcrete based on strength and durability. However, few studies have been conducted on rheological characteristics, which are very important parameters for evaluating the pumpability and shootability of shotcrete. In those studies, silica fume has been generally used as a mineral admixture to simultaneously enhance the strength, durability, pumpability, and shootability of shotcrete. Silica fume is well-known to significantly increase the viscosity of a mixture and to prevent material sliding at the receiving surface when used in shotcrete mixtures. However, the use of silica fume in shotcrete increases the possibility of plastic shrinkage cracking owing to its very high fineness, and further, silica fume increases the cost of manufacturing the shotcrete mixture because of its cost and handling. Colloidal silica is a new material in which nano-silica is dispersed in water, and it could solve the above-mentioned problems. The purpose of this research is to develop high-performance shotcrete with appropriate levels of strength and workability as well as use colloidal silica for normal structures without a tunnel structure. Thereafter, the workability of shotcrete with colloidal silica (2, 3, and 4%) was evaluated with a particle size of 10 nm and silica fume replacement (4 and 7%) of cement. In this study, an air-entraining agent for producing high-performance shotcrete was also used. The rheological properties of fresh shotcrete mixtures were estimated using an ICAR rheometer and the measured rheological parameters such as flow resistance and torque viscosity were correlated with the workability and shootability. More appropriate results will be focusing on the Bingham model properties such that the main focus here is to compare all data using the Bingham model and its performance. The pumpability, shootability, and build-up thickness characteristics were also evaluated for the performance of the shotcrete. This research mainly focuses on the Bingham model for absolute value because it creates an exact linear line in a graphical analysis, which provides more appropriate results for measuring the shotcrete performance rather than ICAR rheometer relative data.

## 1. Introduction

Shotcrete is a method of spraying mortar or concrete with high pressure onto a vertical overhead surface; it is used for surface protection and slope construction [1]. The American Railway Engineering Association (AREA) introduced “shotcrete or sprayed concrete” for the first time in the 1930s. Dry-mix and wet-mix shotcrete are the two types of shotcrete [2].

In this study, wet-mix shotcrete was adopted. Wet-mix shotcrete has good compactness and concrete strength grade with a controlled water–cement ratio and accurate accelerator dosage [3]. Depending on the quality of the concrete before hardening, clogging and pulsation may occur under sustained pressure. Meanwhile, quality control including the water–cement ratio is easily achievable, and the construction capacity is high owing to the high discharge amount [4]. However, quality control is also dependent on the different sizes of the nozzle and hose for adjusting the mix because wet-mix shotcrete pumping performance depends on this [5,6]. The advantage of wet-mix shotcrete is less dust generation and rebound. Shotcrete is mainly used as the primary support material for underground structures and tunnels. It is also used to stabilize a slope. In Europe and North America, shotcrete is used in port and curved-shape structures, as well as the types of structures that are difficult to access [7]. Shotcrete is also applied in the construction and maintenance of complex structures. Korean road construction has maintained the standard of compressive strength is 10 MPa after 24 h, 21 MPa or higher for 28 days of age, and the flexural strength is 4.5 MPa or higher at 28 days of age, and high strength shotcrete has a high strength of 35 MPa or higher at 28 days of age for tunnel shotcrete quality. In Korea, research on shotcrete is mainly conducted on wet-mix shotcrete but nowadays they are also using dry-mix shotcrete. In recent years, the importance of high durability, adhesiveness, compatibility, and strength of concrete verification of the constructability, which is represented by workability and shootability, been investigated through various studies such as rheology [8]. There are some verified examples such as “High-strength shotcrete by reducing the water-cement ratio of wet-mix shotcrete formulation to 40%, additionally, since the early 1990s, Morgan et al. have actively implemented addition of silica fume in shotcrete.” From the 1990′s researchers successfully used the durability of shotcrete on the road engineering which has been actively conducted” [9]. Moreover, Chen [6] et al. examined the rheological parameters, slump, and air content of the fresh concrete, which were evaluated before and after pumping, respectively. Additionally, this study indicated the double-plunger pump system and lubrication effect of cement paste were related to the operation, and drastically change concrete properties such as flowability and torque viscosity. K.K. Yun et al. [10] investigated rheological properties such as shootability and pumpability of the high-performance wet-mix shotcrete with silica fume, an air-entraining agent, superplasticizer, synthetic fiber, powder polymer, and a viscosity agent. Among the major findings of this study, they found a superplasticizer had a relatively greater impact on the flow resistance than the torque viscosity. Moreover, it was observed that silica fume led to an exceptional enhancement in flow resistance while slightly reducing torque viscosity. This behavior trend indicates that silica fume is quite effective in improving the rheological properties of HPWMS, particularly in terms of shootability and pumpability. Choi et al. [11] measured the laboratory evaluations, which were conducted to quantify the effect of air-entraining agents and silica-fume on the air void characteristics of wet-mix shotcrete before and after the shotcreting process with permeability properties. Moreover, the major finding of this study was the decreased overall air content in wet-mix shotcrete in the shotcreting procedure; 4.5% silica fume replacement, which was incorporated with AEA, confirms both an adequate spacing factor and good retention of small entrained air bubbles even after shotcreting, which may upgrade the freeze–thaw and scaling resistance. The air content affected the permeability of WMS, but no consistent correlation was found between the spacing factor and permeability.

Silica fume is frequently used as a blend material for shotcrete. Silica fume is a miscible material that generally uses different percentages in the concrete by replacing cement to form a homogenous mixture when added together. It has high fineness with a particle size of approximately 0.1–0.3 µm [12]; thus, it is possible to secure high strength and durability owing to an increase in water tightness and a pozzolanic reaction. In particular, when used in shotcrete, silica fume can increase the viscosity of concrete and cause the clogging phenomenon, which is a material-sliding phenomenon occurring on the pouring surface [13]. It is the best blending material that can improve the viscosity and mixing quality of the concrete. Based on these studies, silica fume can secure durability, strength, and ensure pumping as well as shooting properties simultaneously. Colloidal silica is also used in shotcrete for better workability performance due to being finer than silica fume with more surface area.

This study attempts to examine the silica fume and colloidal silica that are used as part of wet-mix high-performance shotcrete. In this study, silica fume and colloidal silica are used to understand the rheological properties of high-performance shotcrete. Additionally, colloidal silica (0, 2, 3, and 4%) variables and silica fume (4 and 7%) are used for the purposes of this study. Herein, we also briefly described the step-by-step test process of the analysis used in this study.

Expected workability is measured by the rheology characteristic using a rheometer. The rheological analysis is used for measuring the concrete workability, plastic viscosity, flowability, etc. [14]. Silica fume and colloidal silica are used for improving the performance of high-performance shotcrete based on the rheology perspective. Herein, the comparative analysis results for the performance of shotcrete between the silica fume and colloidal silica mixture are exhibited. When mixing the different amounts of colloidal silica, H (torque viscosity) is decreased and G (flow resistance) exhibits an increasing tendency [15].

The main focus of this research is the rheological behavior observed based on IBB and ICAR rheometers. However, these values depend on the rheometer parameter and model. For the exact and absolute value, the rheological data change because of the conversion of torque viscosity and flow resistance to plastic viscosity and yield stress, respectively. Moreover, the Bingham model shows the exact linear (R^2^) line by the regression formula. So, Bingham model analysis is the main novelty in this study.

## 2. Experimental Materials and Formulation

In this study, the materials used are ordinary Portland cement, sand, and 10 mm crushed aggregate. High-performance shotcrete is used with admixtures such as silica fume and colloidal silica with different percentages based on the cement amount and superplasticizer as well as the air entraining agent. 

### 2.1. Cement 

Ordinary Portland cement (Sungshin Cement Industry, Korea) was used in this experiment. The cement’s main chemical composition is 61.2% CaO, 20.8% SiO_2,_ 6.3% Al_2_O_3_, 2.3% SO_3_, and 3.3% MgO. The specific area (cm^2^/g) and density (g/cm^3^) of the cement used are 3300 cm^2^/g and 3.15 g/cm^3^ accordingly.

### 2.2. Aggregates

Coarse aggregate with maximum dimensions of 10 mm for the stone and fine aggregate (river sand) were used in this study. Coarse aggregate and fine aggregate satisfied the Korean standard (KS) with the requirements of the experiment. The coarse aggregate absorption rate and density are 1.95% and 2.69 g/cm^3^ (SSD). The particle size of the mixed aggregate for shotcrete satisfies the Korea Expressway Corporation’s shotcrete quality standards, and mixed aggregates with a fine aggregate ratio of 75% are used. According to the Korea Expressway Corporation, shotcrete technology follows a 65% sand-to-aggregate ratio for tunnel shotcrete. However, this study used a 75% sand-to-aggregate ratio considering the normal structural surface finishing and rebound reducing. The fineness moduli of the coarse and fine aggregates are 5.88 and 2.82, which are based on the particle size distribution of the aggregate. Therefore, we can conclude that it is a well-graded mixed aggregate.

### 2.3. Admixtures

#### 2.3.1. Silica Fume

Silica fume, also known as micro-silica, is a consequence of manufacturing alloys as well as elemental silicon in an electric arc furnace. During the process of producing silicon metal or ferrosilicon alloy using coke, quartz is obtained through the reduction. The main component is silica, which is approximately 97% of the total components; mean particle size of 0.15 µm of the spherical is considered a fine powder. 

Silica fume is a spherical shape as a fine powder, which is used as a concrete micro filler effect and pozzolanic reaction in concrete [16]. Due to the use of silica fume, the transition region decreases, and the characteristics are improved through the very dense structures formed. Moreover, the concrete strength, durability, and adhesion are dependent on the silica fume (SF) mixes incorporated with the concrete. This promotes several advantages of shotcrete properties and an increase in the discharge ability of a mixing material used in the experiment [17]. Using the generalization, silica fume common usage in concrete at the weight of 3~5% is reported, but according to the Commission Delegated Regulation (EU), 0~10% is encouraged within concrete [18]. It can be divided into a filling effect and a pozzolanic reaction, with calcium hydroxide produced through a hydration reaction when the pore water inside accumulates inside the transition region between the paste and aggregate particles. Moreover, 4 and 7% silica fume are used in the research experiment.

#### 2.3.2. Colloidal Silica

Colloidal silica has a negative charge strip in amorphous silica (SiO_2_), which constitutes fine particles in colloidal water and is apparently a totally transparent or translucent white shade in a solution of water. In a dispersion medium, it is distributed to the shape of colloidal silica, which implies partial stabilizers including a state of colloidal silica. If Colloidal silica’s fine particles are 7 nm or smaller, then consequently, colloidal silica’s appearance is transparent or similar to water; 10~30 nm is another transparent case, and an approximately 50 nm particle size becomes a white, milky color which are shown in Figure 1. In addition, a particle size of 10 nm for colloidal silica was selected, and 2, 3, and 4% were proposed as appropriate mixing amounts.

In colloidal silica, the silica particles are present on the surface of the alkali -OH, and these alkali ions are formed by the double electric layer (electric double layer) [19]. Negatively charged particles between the repulsive force occur due to colloidal silica, which is maintained as chemically and structurally stable conditions. The electrochemical stability in the state charge balance is maintained to balance the break particles to each entanglement and the viscosity increases the gelling and aggregation, as the reaction takes place [17]. According to previous studies, the addition of colloidal silica improves the performance of cement mortar and concrete. For colloidal silica, a liquid-type admixture with 40% solid content of colloidal silica was adjusted and designed.

#### 2.3.3. Air Entraining Water Reducing Agent

A high-performance air-entraining water-reducing (AEWR) agent was used in this study, and the AE agent is used at 0.2% in the mixture to ensure the slump and workability of the high-performance shotcrete. Furthermore, the physical properties of the high-performance AEWR agent are explained in Table 1.

### 2.4. Experimental Mix Design and Formulation

This study attempted to analyze the performance of the shotcrete based on different mixing rates of all ingredients. Moreover, the mixture designations were named silica fume and colloidal silica and the mixing percentage was replaced by 460 kg/m^3^ of cement, while a sand-to-aggregate ratio of 75% and a water–cement ratio of 40% were used in this study. The target slump was 120 ± 30 mm. In order to be satisfied, the amount of high-performance AEWR agent was adjusted for the air content in the mix, at 7 ± 2%, and shotcrete performance. The colloidal silica mixing ratio was selected as 0, 2, 3, and 4% for this study. Colloidal silica is incorporated with different percentages of the high-performing AE water-reducing agent to secure the target slump. For cement hydration, the optimum mixing ratio of silica fume for calcium hydroxide and the pozzolanic reaction is 7%. For a comparison of the same mixing ratio with silica, a 4% mixing ratio variable was selected. Table 2 shows the experimental variables.

## 3. Experimental Process and Analysis Methods

### 3.1. Shotcreting Method and Testing

This study focuses on the wet-mix shotcrete process. Wet-mix shotcrete is produced by batch plant production, and the shotcrete mixture is transferred after mixing in the plant for shooting.

Using this process mainly involves various types of equipment that are described herein [20]. First, we used a piston-type shotcrete pump and subsequently used an air compressor, which has a FAD value of 23.3/819 m^3^/min/scfm. Preparation of the test specimens is very important because the nature of the shotcrete is slightly different such that it is not possible to produce test specimens that are directly poured into the circular mold. The shooting hose and nozzle diameter in this test were 2 in and the test panel was situated 1.5 m from the nozzle. Here, the shooting performed for the shotcrete test is shown in Figure 2.

### 3.2. Slump and Air Volume Tests

Korean standard KSF 2402 is based on the slump test method of concrete that was performed on fresh concrete for determining the properties of the mixture. Moreover, KSF 2421 is used to determine the air-entrained air amount in the concrete. For example, in this study, the target slump was 120 ± 30 mm, and the air volume before shotcrete placement was 8.0 ± 1.5%. The mix design was performed by fulfilling the required criteria that satisfied the purpose of using the shotcrete. In addition, after shooting, an air volume test was performed to ensure that the air volume of the material was measured perfectly, which is shown in Table 3.

### 3.3. ICAR Rheometer and Bingham Model

There are a few rheometers, such as IBB, ICAR, and Btrheom rheometers. Among these, the ICAR rheometer was used in this study. The ICAR rheometer was first evolved as a rheology measurement from the University of Texas in Austin and thereafter produced by a German company in Denmark for the flow behavior of concrete mixtures. This test device mainly includes or consists of a container that can hold concrete before hardening, an electric motor, a torque meter, a four-blade vane that is held by the chuck on the driver, a frame to attach the driver/vane assembly to the top of the container, attaced containers and head equipment that send measurement data in real-time to a computer, and finally, a program to analyze the test data. In the real-time record, the test apparatus is used to evaluate the yield stress and plastic viscosity for flow curves [15]. Following that, the Bingham parameters are automatically calculated on the basis of those rheological data. As shown in the graph, rotational velocity is signified by the X-axis, while torque values are indicated by the Y-axis. The values of the descending rotational vane speed and torque are demonstrated and linearly assessed to determine the concrete’s rheological behavior. Figure 3 and Figure 4 depict the ICAR Rheometer and the cylinder center axis.

Figure 5 and Figure 6 show the equipment and possible evaluation of the results used in the experiment. The test equipment can perform two types of tests. First, the rotation speed is 0.025 rev/s, which is indeed a stress growth test that runs at a consistent moderate speed to build up torque and is measured as a function of time. The maximum torque measured during the test is used to calculate the static yield stress. Another type of test is a flow curve test to determine the dynamic yield stress and plastic viscosity. For the test method, after rotating the impeller at the maximum speed, the speed decreases, and measuring the number of torques is the basic part of the test process. It stores the measured torque and impeller speed in real-time, and the calculation of Bingham parameters is performed automatically. The main equation of the ICAR rheometer is as follows:**T = G + HN**(1)

**T** = Torque.

**G** = Flow resistance (Nm).

**H** = Viscosity factor (N.m/rps).

**N** = Rotational Speed. (rev.s^−1^).

However, the exact or absolute value is determined by the Bingham equation below.
τ = τ_o_ + µγ(2)

τ = Shear stress (Pa).

τ_o_ = Yield stress (Pa).

µ = Plastic Viscosity (Pa.s).

$\dot{\gamma \;}\;$= Shear rate (per sec).

The calculation of yield stress and plastic viscosity, on the other hand, is used as a recursive nonlinear optimization method in the Bingham model from the ICAR rheometer. If the conventional Reiner–Riwlin formula has been used in the presence of a dead zone, the error in the quantified rheological parameters can be noteworthy. The range of shear stresses existing in the annulus of a coaxial-cylinder rheometer may not be sufficient to cause all material to flow as fluids with yield stress. As a result, there is no flow in the dead zone. The appearance of a dead zone is repeatedly alluded to as plug flow in the concrete literature, but this term is a slight misnomer because the problem with plug flow is that not all of the material flows. The term plug flow is used in general rheology to describe the movement of yield stress fluids through a pipe [21]. This equation is provided as follows [22]:(3)$$\Omega =\frac{T}{4\pi h\mu }\left(\frac{1}{R_{1}^{2}}-\frac{2\pi h\tau_{0}}{T}\right)-\frac{\tau_{0}}{2\mu }\ln\left(\frac{T}{2\pi h\tau_{0}R_{1}^{2}}\right)$$

Ω = Angular velocity.

*R*_1_ = Inner cylinder radius.

*R*_2_ = Outer cylinder radius.

*h* = Cylinder height.

*T* = Torque.

*τ*_0_ = Yield stress.

*µ* = Plastic viscosity.

The final experimental analysis, as seen in Figure 6, demonstrates the rotational speed on the X-axis as well as the torque value on the Y-axis. The graph depicts the rotational speed and torque value at which the speed decreases, as well as a linear analysis. At this point, the Y-axis intercept represents the flow resistance (G) value, which is converted to the yield stress (Nm) in the Bingham model, whereas the inverse of the linear slope represents the plastic viscosity (Nm.s) value in the Bingham model. The lower the plastic viscosity, the better the pumpability of shotcrete, and the higher the yield stress, the thicker the paste after pouring the shotcrete.

## 4. Test Results and Analysis

### 4.1. ICAR Rheology According to the Ratio of Colloidal Silica and Silica Fume

The effect of the mixing ratio on the workability of shotcrete was determined in this study. Here, the volume fractions of colloidal silica are 2%, 3%, and 4%. The variable to be incorporated was selected during which rheological characteristics were identified. This is considering the shotcrete pumping and sticking properties of the colloidal silica.

Figure 7 shows the influence of the dosage of colloidal silica. It is a graph that linearly analyzes the results of the rheology measurement. The linearity is located vertically to the OPC variable, and the highest dosage of the colloidal silica (CS4.10) variable is located at the highest vertical distance followed by CS3.10 and CS2.10. In addition, the linear slope of the OPC variable was the highest, followed by CS2.10, CS3.10, and CS4.10. This happened because silica particles dispersed in colloidal silica are mixed with concrete. This can be observed due to the decreasing the viscosity of concrete. However, there is one problem when describing the results because the points of the torque and speed are not perfectly linear, which means the R^2^ value of these are varied, whereby the CS2.10 and CS3.10 values are 0.9986 and 0.9972, respectively. Creating the linear line requires the linear regression line that is shown in Figure 7. Therefore, the Bingham analysis is more effective. In the Bingham model, the lines are linear and depend on the shear stress and shear rate of the concrete. Here, we can determine the plastic viscosity instead of torque viscosity as shown in Figure 8. Additionally, the behavior of the results is the same in the rheological data. Therefore, we can consider the Bingham model data. The CS4.10 shear stress (τ) Pa is higher than other mixes, respectively.

Concrete incorporated with silica fume significantly improves viscosity, pumpability, and adhesion. Due to the fineness of silica fume, the durability of internal charging is increased, and the long-term strength is increased by the pozzolanic reaction of the concrete. Furthermore, 4% and 7% silica fume was selected as variables, represented by SF4 and SF7. The results of rheological properties according to the mixing of silica fume are compared to the CS4.10 variables that incorporated colloidal silica. This mixing uses the same mixing ratio to evaluate the material performance of silica.

Figure 9 and Figure 10 show a linear analysis of the rheology test results and the Bingham model test results according to the mixing rate of silica fume. These graphs are shifted vertically according to the incorporation of silica fume. The SF7 variable is located on the upper vertical side of the SF4 variable; when the amount of silica fume increases, the yield stress can be predicted to increase. The slope of the SF7 variable with 7% silica fume tends to be larger than the SF4 variable with silica fume mixed at 4%. The decrease in torque viscosity and plastic viscosity was predicted as the silica fume mixing ratio increased. Figure 9 and Figure 10 show the comparison of silica fume dosage and colloidal silica mixing dosage. The rheological data of CS4.10 show the linearity of the variable, which is compared with silica fume. It can be observed that it is similar to the variable and the slope is similar to the SF4 and SF7 variables.

Table 3 shows the amount of slump and air before shooting according to the change in the mixing ratio of colloidal silica and silica fume. It shows the test result of the rheology and the Bingham model test results. A target slump of 120 ± 30 mm for all experimental variables and an air volume of 7 ± 2% were satisfied for better performance of the shotcrete. Torque viscosity H from OPC is 8.70 Nm.s, which is measured to have the highest torque viscosity; CS2.10, CS3.10, and CS4.10 torque viscosity were measured accordingly as 6.65 Nm.s, 6.15 Nm.s, and 5.50 Nm.s. Meanwhile, in OPC, flow resistance G was measured to be 1.45 Nm, which was the lowest flow resistance. It was measured to be 2.87 Nm in CS2.10, 3.06 Nm in CS3.10, and 3.61 Nm in CS4.10. Based on the Bingham model, the results also exhibited the same behavior as the rheometer results. The value of the plastic viscosity (µ) of the OPC is 170.6 Pa.s and it is the highest value. Meanwhile, the yield shear stress (τ_o_) of the OPC is the lowest value in all mixes at 228.2 Pa. Herein, we also show that the increasing percentage of the colloidal silica and silica fume cause remarkable changes to the plastic viscosity and shear stress. The plastic viscosity is gradually decreased, implying that it is more workable and more pumpable, and yield shear stress is increased gradually, thereby initiating the initial flow with good flowability.

Additionally, OPC’s torque viscosity (H) is 8.70 Nm.s, which is the highest compared to other experiments, and according to Table 3, the torque viscosity (H) of SF4 and SF7 is 5.54 and 4.56 Nm.s,. which shows the silica fume incorporation rate is low as it increased. Further, the torque viscosity (H) value of SF4 (5.54 Nm.s.) is somewhat similar to the value of the incorporated amount of colloidal silica CS4.10 (3.48 Nm.s). Flow resistance (G) has an SF4 of 2.75 Nm and SF7 of 3.48 Nm, which increases the silica fume mixing rate. It was measured to be high. The flow resistance of 3.48 Nm in the SF7 variable is similar to CS4.10.

Figure 11 graphically shows the analysis result of the torque viscosity (H) in all experimental mixing ratios. With OPC as a reference variable, the torque viscosity decreased by 24% in CS2.10 and decreased by 37% in CS4.10. It was confirmed that the torque viscosity (H) decreased as the mixing ratio increased. Herein, we attempt to briefly describe the plastic viscosity using the Bingham model. The Bingham model shows the analysis result for plastic viscosity (µ). As per the reference variable OPC, the plastic viscosity decreased by 40% in CS2.10 and decreased by 60% in CS4.10. Therefore, the same behavior is also shown in the Bingham model in Figure 12. Figure 11 shows the analysis result of torque viscosity H according to the silica fume mixing ratio. Torque viscosity of SF4 decreased by 36% compared to OPC, and SF7 decreased by 48%, resulting in silica. It is considered that a high level of the pumping property can be secured when silica fume is incorporated with concrete. The Bingham model result also achieved the same behavior with more usable data because of the more effective linear data.

Figure 13 presents a graph of the analysis result for flow resistance (G). Based on the OPC variable, CS2.10 is 198%, CS3.10 increased by 211%, CS4.10 was measured as 249%, and as the mixing ratio increased, the flow resistance G tended to be high. When we used the Bingham model, this behavior of the yield stress was similar because the result difference was excessively high such that it focused on the relative yield stress of CS2.10, which was 256.9% based on OPC. Here, the mixing proportion increased, and the relative yield stress also increased respectively, such that CS 3.10 was 286.2% and CS.4.10 was 360.7%.

Thereafter, silica fume mixing also exhibited the same behavior; SF4 was 248.6% and SF7 was 350.6%, as shown in Figure 14. The pumping and sticking properties are measured using the measured torque viscosity H and flow resistance G. The appropriate dosage was selected through analysis. As the mixing ratio of colloidal silica increased, the torque viscosity was small; therefore, the pumping property improved, and the flow resistance increased such that the material adhesion increased after shooting. We believe that it will improve the proper maintenance of mixing for shotcrete and high-performance concrete. To achieve a high-performance target slump, a higher mixing ratio of 4% incorporation (CS4.10) was selected as the appropriate incorporation rate because of the fear of overuse of the AE water-reducing agent.

### 4.2. Comparison with the Results of IBB Rheometer

The IBB rheometer is a non-coaxial cylinder, H-shaped impeller popular rheometer for concrete rheology measurement. The main remarkable property of the IBB and ICAR rheometers is their similar measurement characteristics, such as G and H. However, the value of the torque viscosity and flow resistance are not equal because of the dimensional measurement of the rheometer. Therefore, we used Jeon’s data with an IBB rheometer to show the comparison [23] with the ICAR rheometer. In Jeon’s mixing proportions, the author mainly used SF5%, SF10%, and SF15%. This is slightly different from our research mixing, but we can observe the behavioral change on the plastic viscosity and yield stress of the Bingham model.

Figure 15 shows the result of the rheological characteristics based on reference (REF) and it was observed that the SF5%, SF10%, and SF15% flow resistances were gradually vertically increased, and torque viscosity also gradually decreased with the mixing proportion instead of SF15%. Meanwhile, the Bingham model shows the same type of behavior for yield stress and plastic viscosity as in Figure 16. As per our research results, Dr. Jeon’s data exhibit similar characteristics. Table 4 shows Jeon’s rheology and the Bingham model characteristics according to the silica fume mixing ratio.

Torque viscosity (H) from REF is 2.59 Nm.s, which is measured using the IBB rheometer and SF5, SF10, and SF15 torque viscosity H were measured accordingly as 2.27 Nm.s, 2.1973 Nm.s, and 3.23 Nm.s. Meanwhile, in RFE, the flow resistance (G) was measured to be 5.5904 Nm. It was measured to be 5.49 Nm in SF5, 6.988 Nm in SF10, and 10.822 Nm in SF15. Based on the Bingham model, the results also showed the same behavior. Herein, we also observe that the increasing percentage of silica fume caused a remarkable change in the plastic viscosity and shear stress. Plastic viscosity gradually decreased, implying that it is more workable and more pumpable; however, yield shear stress increased gradually, leading to the initiation of the initial flow with good flowability. The only remarkable behavior was exhibited by SF15 because its plastic viscosity and yield stress are higher than those of the others. Herein, we also need to explain that our test results and Jeon’s results have some significant differences due to the rheometer. However, when these data changed in the Bingham model, the behavior of the data remained the same. Therefore, the Bingham model conversion is more effective than the rheometer data.

Figure 17 and Figure 18 graphically show the analysis results for torque viscosity H and plastic viscosity µ. The torque viscosity decreased by 12.4% and 15.32% according to SF5 and SF10, respectively. It was confirmed that the torque viscosity H decreased as the mixing ratio increased. This was also observed in our research. However, there is more to note in SF15. Its behavior is completely different from others; its torque viscosity spontaneously increased to 124.52%. The results of the Bingham model’s plastic viscosity are the same as the rheology data. Meanwhile, the flow resistance, G, and yield stress gradually increased as per the proper behavior such that SF5 to SF10 is 26.78% and SF15 is 95.36% according to the flow resistance, which is shown in Figure 19 and Figure 20. Moreover, yield stresses saw the same type of increase as per the increasing mixing ratio.

Thereafter, all results are combined in one graph as shown in Figure 21. The graph mainly shows all the Bingham model behaviors in one figure. We can observe the exact behaviors of all mixes from this figure.

### 4.3. Pumping Properties

Shotcrete’s pumping properties are mainly dependent on the material rheological properties, equipment performance (pumping rate of 4.6 m^3^/h), and hose diameter of the shotcrete technology. In this study, the rheological characteristics were measured using an ICAR rheometer with a coaxial vane. The size of the drum is reflected in the measurement of the shotcrete torque as well as the distance to the wall. Therefore, the measurement factors are different for every rheometer. Figure 22 shows research data and Jeon’s data. However, Jeon used an IBB rheometer while an ICAR rheometer is used in this research to measurine the rheological parameters. However, for more effective results and comparisons, the Bingham Model was used. According to the theory of pumpability, the results depend on a good combination of yield stress and plastic viscosity. Besides that, enough fines in the mix must build up a lubrification layer, and the stability of the mix against segregation has to be tested. In this paper, only the basic rheological behavior of the mixes is investigated. Based on the pumpability characteristics, plastic viscosity decreased, then the pumping ability of the shotcrete increased. Therefore, OPC has poor pumpability because the plastic viscosity (170.6 Pa.s) and yield stress (228.2 Pa) combination is not good for pumping. Moreover, SF15 has some problems because the gap in the results is more, such as plastic viscosity (49. 11 Pa.s) and yield stress (1471.79 Pa). Besides, all the other combinations have good pumpability based on the plastic viscosity and yield stress combination. Figure 22 shows that CS2.10, CS3.10, CS4.10, SF4, SF5, SF7, and SF10 have pumpable capacity in the shotcrete process. Meanwhile, if yield stress is increased, then the adhesion quality after shooting is also increased. Moreover, Figure 22 indicates the possible recommended area (or workability box) for good pumpability with the mixes and equipment used in this research. However, the precondition in this research was enough fines for a sufficient lubrification layer and no segregation.

### 4.4. Build-up Thickness

Beaupre (1994) [24] proposed the thickness of shotcrete according to the flow resistance (G). To investigate the possible build-up thickness of the different mixes, a spraying test was performed with the equipment described in Figure 2 on a vertical wall. No accelerator was used. Herein, we mainly show the relationship between the build-up thickness and yield stress because, as per the previous description, we know that the yield stress of the Bingham model has a relation to the flow resistance of the rheological data. Table 5 lists the thickness of paste based on Figure 23. The variable mixing of colloidal silica CS4.10 was applied to a thickness of 213.1 mm. Additionally, between SF4 and SF7, SF7 is more adhesive because of the build-up thickness of 204.1 mm. Meanwhile, the OPC build-up thickness of the paste is predicted to be 99 mm; thus, only the thickness of 100 mm is not secured according to Beaupre (1994). Here, we attempt to create the relationship between the yield stress and buildup thickness by drawing a polynomial line of the points, which is shown in Figure 23. From Figure 23, the points of the buildup thickness that are related to yield stress are very close. Therefore, this combination is effective to show the relationship.

## 5. Conclusions

In this study, the main concern is the rheological properties based on the Bingham model data of high-performance wet-mix shotcrete mixed with colloidal silica. Colloidal silica dosages of 2, 3, and 4% and the experimental variable particle size of 10 nm were selected, which are typically used for existing high-performance shotcrete. Additionally, silica fumes were compared with dosages of 4 and 7%. Measured by rheology tests, the data are analyzed for pumping and adhesive properties of shotcrete. From the results, the following conclusions were drawn:As the dosage of colloidal silica increases, the torque viscosity H decreases, and the flow resistance G increases. In addition, we transformed the data to absolute rheological values through the software from the rheometer producer, which also exhibited the same characteristics for the plastic viscosity µ and yield stress τ_o_. That data can be used to compare results with results from other rheometers with other geometry. When producing high-performance shotcrete using 4% colloidal silica particles, shotcrete with improved pumpability and adhesion production is expected to be possible.As the mixing rate of silica fume increases, the torque viscosity H and the plastic viscosity µ decrease, whereas the flow resistance G and yield stress τ_o_ increase. It seems to improve the functional behavior of shotcrete. As a result of the comparative analysis, the same amount of colloidal silica was used as silica fume. The results indicate that the use of colloidal silica would be more beneficial for obtaining a higher level of shootability.As the colloidal silica dosage is increased, the build-up thickness is also increased gradually and the same behavior for the silica fume percentage is observed. This relationship is shown only with yield stress because it has no relationship with plastic viscosity.A different amount of the AE water-reducing agent was used to satisfy the same slump when incorporating colloidal silica with high-performance shotcrete. Additionally, the size of the colloidal silica particles is indirectly inversely proportional to the amount of target slump required to secure it, and the size of the colloidal silica particles showed a tendency to increase the amount of high-performance AE water-reducing agents. Moreover, it was evaluated that the dosage of colloidal silica increased, then viscosity decreased and flowability decreased gradually.


Rheological properties of high-performance shotcrete mixed with colloidal silica and silica fume were the major finding in this study, which were used to analyze and evaluate the shotcrete performance of shootability and workability. In this study, Bingham model transformation with ICAR data conveys great value. In the future, we can use different models such as the modified Bingham model, Herschel_bulkley, etc., for comparison studies. Based on the rheological properties, this study impacts the civil engineering construction field. Colloidal silica is less applied to general concrete. Moreover, in future aspects, we can use the different particle size colloidal silica to show a better performance of shotcrete. In the future, it will be necessary to analyze the various durability characteristics through the durability test of shotcrete for better performance. Moreover, in future, real pumping tests recording the resulting pumping pressure as well as tests on the lubrication layer of the investigated mixes could complement these investigations.

## Figures and Tables

**Figure 1 materials-15-00428-f001:**
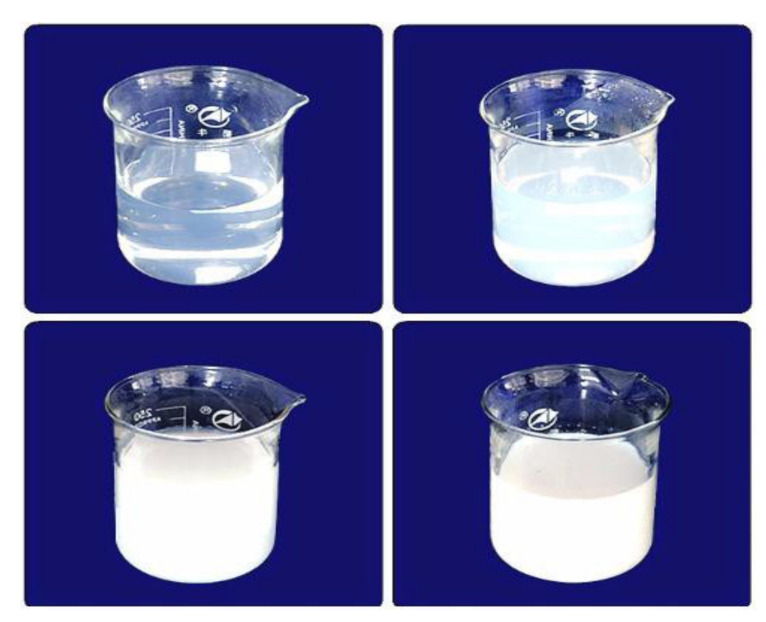
Color according to the size of colloidal silica particles.

**Figure 2 materials-15-00428-f002:**
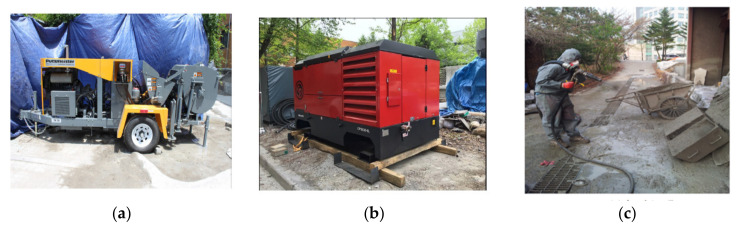
(**a**) Shotcrete equipment with mixer, (**b**) air compressor, (**c**) shooting in the panel, shotcreting method.

**Figure 3 materials-15-00428-f003:**
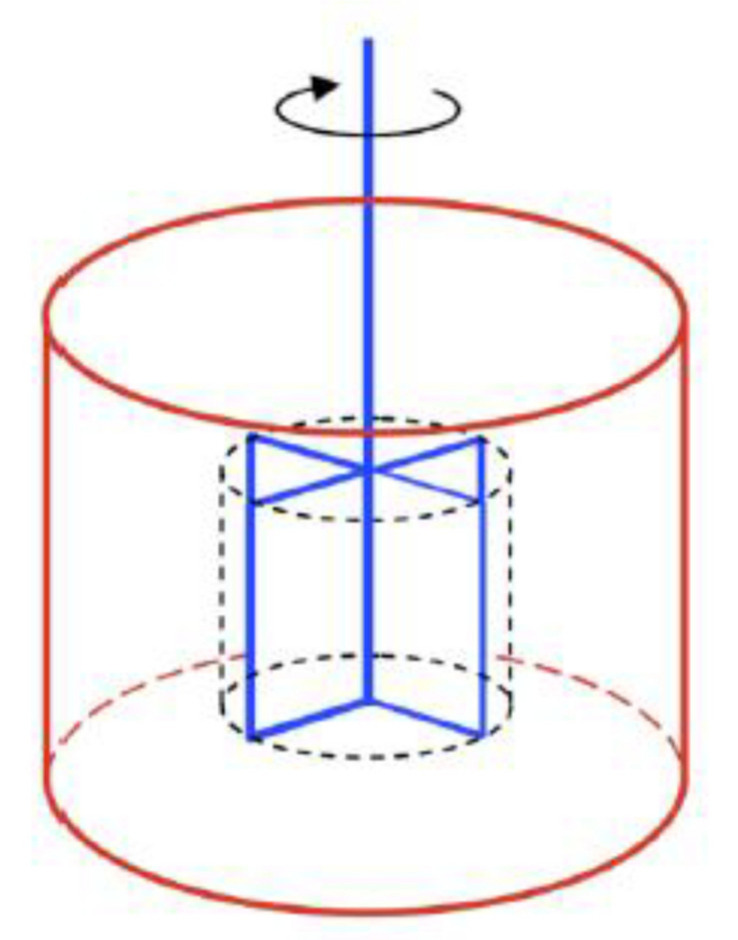
Central shape of ICAR rheometer [15].

**Figure 4 materials-15-00428-f004:**
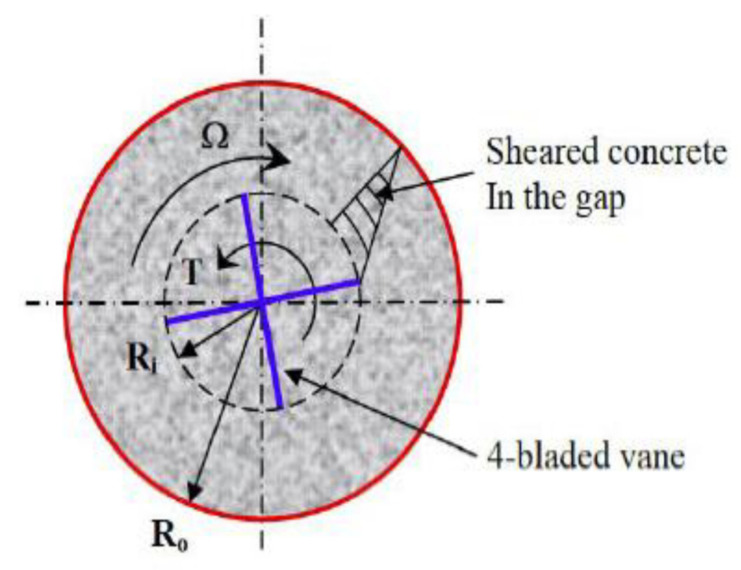
Test principle of ICAR rheometer [15].

**Figure 5 materials-15-00428-f005:**
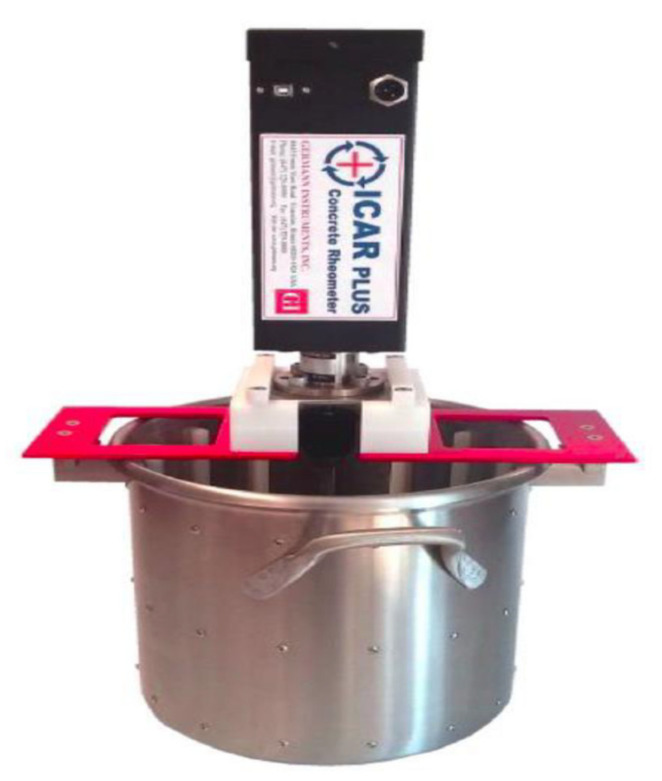
ICAR rheometer. The parameter and rheometer models influence this equation.

**Figure 6 materials-15-00428-f006:**
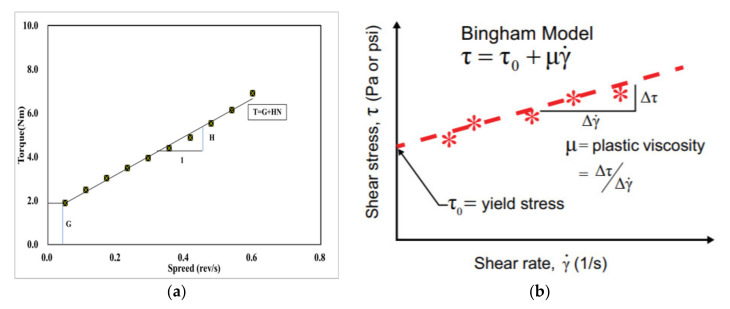
(**a**) Relative values for the ICAR rheometer, (**b**) absolute values calculated by solver of ICAR-rheometer, example of the concrete rheology test results using ICAR rheometer.

**Figure 7 materials-15-00428-f007:**
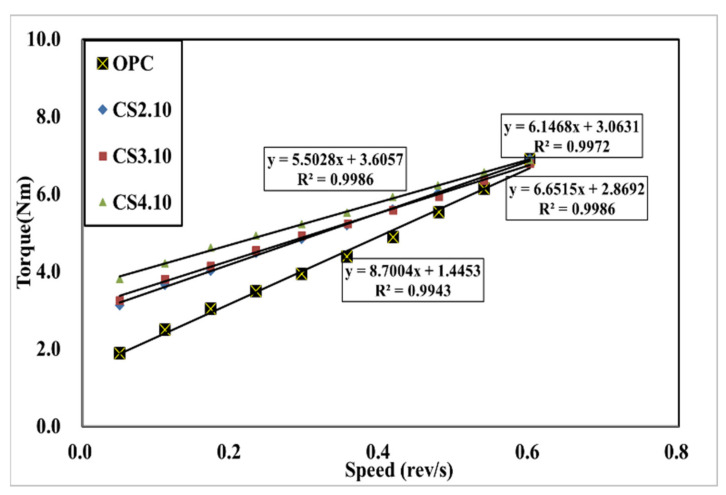
Rheological measurement results according to the mixing ratio of colloidal silica.

**Figure 8 materials-15-00428-f008:**
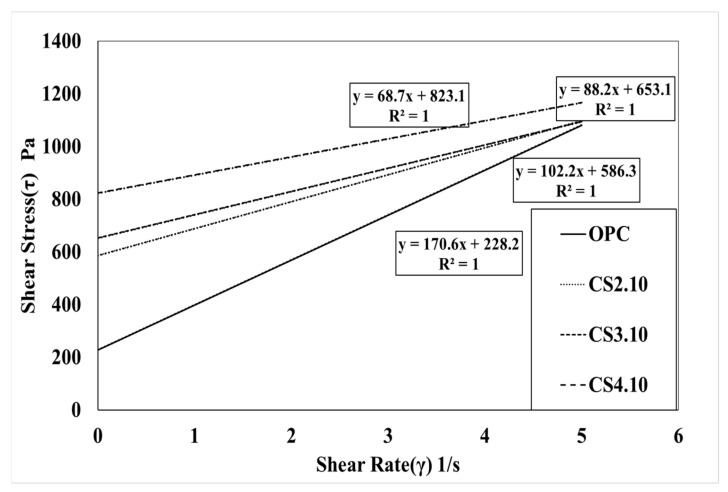
Bingham model results according to the mixing ratio of colloidal silica.

**Figure 9 materials-15-00428-f009:**
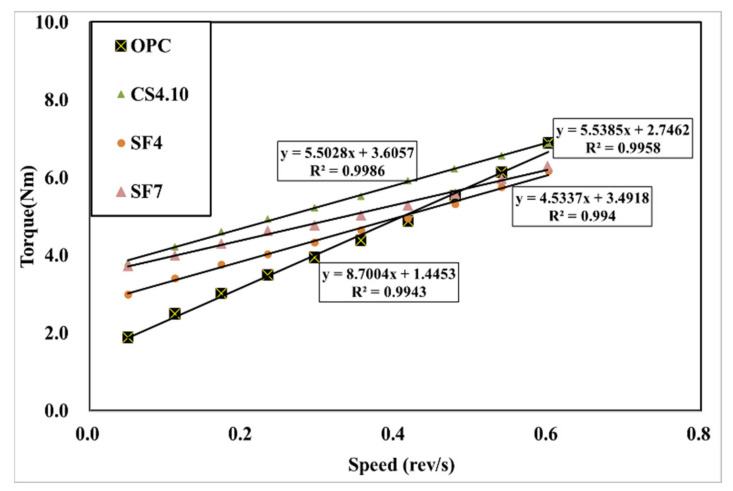
Rheological measurement results according to the mixing ratio of silica fume and colloidal silica.

**Figure 10 materials-15-00428-f010:**
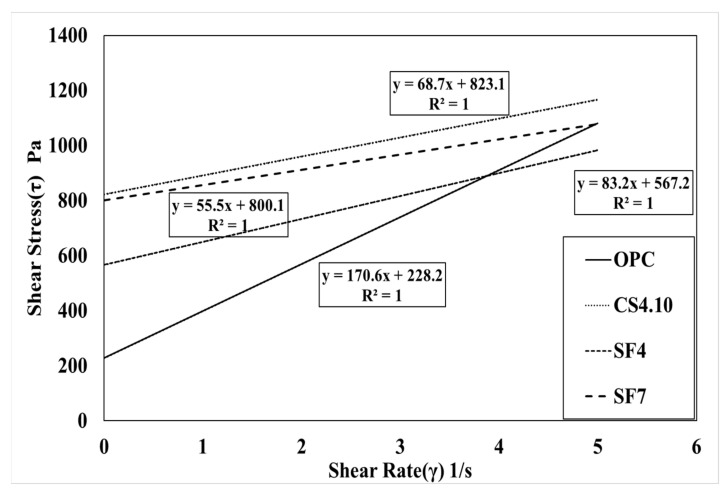
Bingham model results according to the mixing ratio of silica fume and colloidal silica.

**Figure 11 materials-15-00428-f011:**
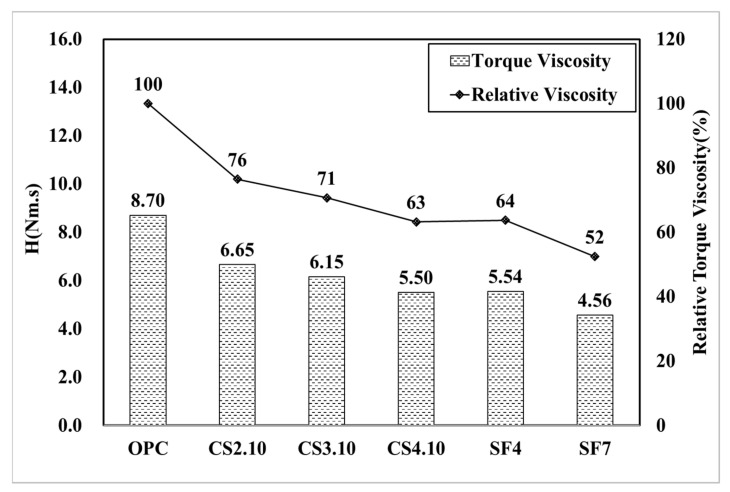
Torque viscosity (H) of the mixing ratio.

**Figure 12 materials-15-00428-f012:**
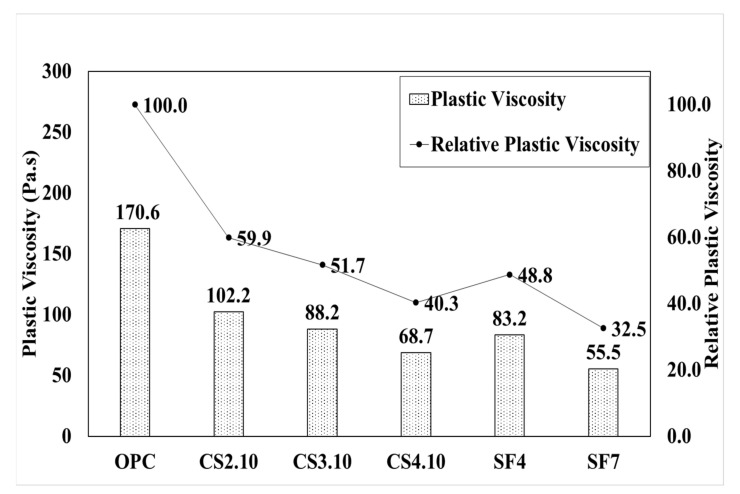
Plastic viscosity (µ) of the mixing ratio.

**Figure 13 materials-15-00428-f013:**
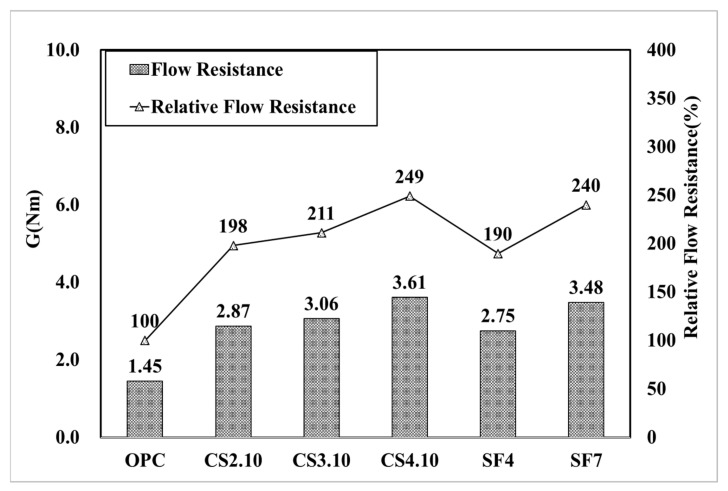
Flow resistance (G) of the mixing ratio.

**Figure 14 materials-15-00428-f014:**
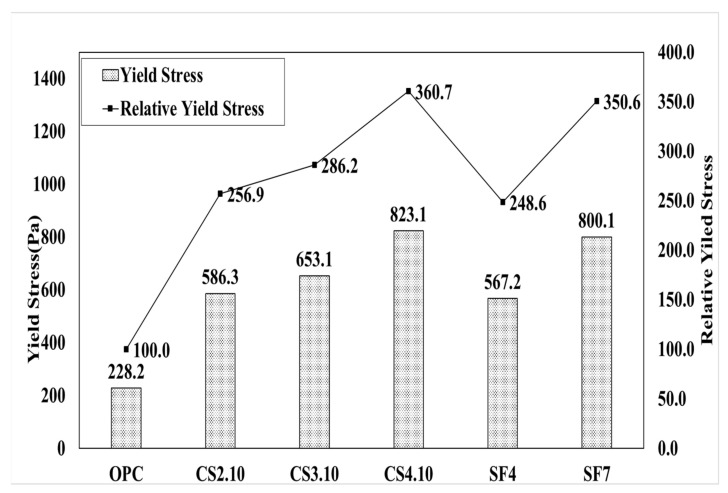
Yield stress (τ_0_) of the mixing ratio.

**Figure 15 materials-15-00428-f015:**
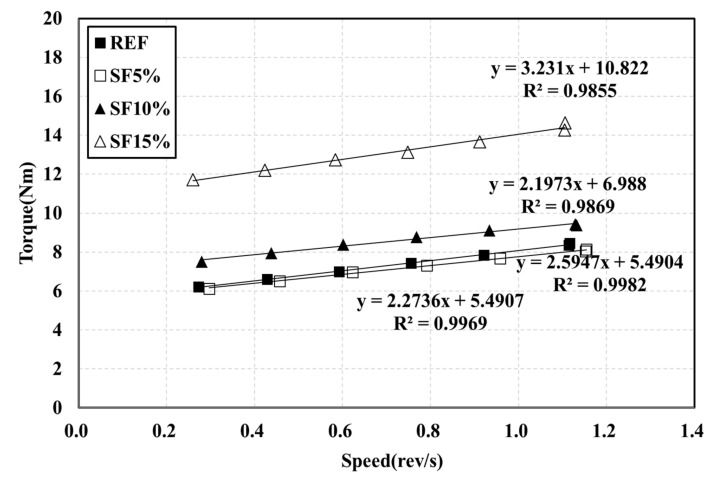
Rheological measurement results according to Jeon’s mixing ratio.

**Figure 16 materials-15-00428-f016:**
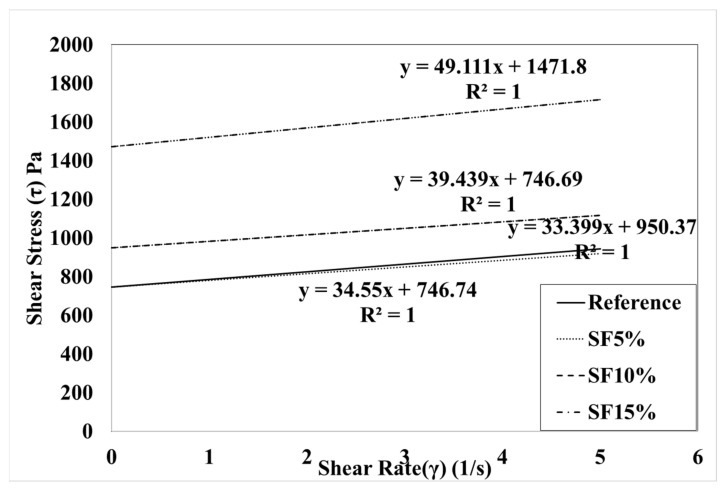
Bingham model results according to Jeon’s mixing ratio.

**Figure 17 materials-15-00428-f017:**
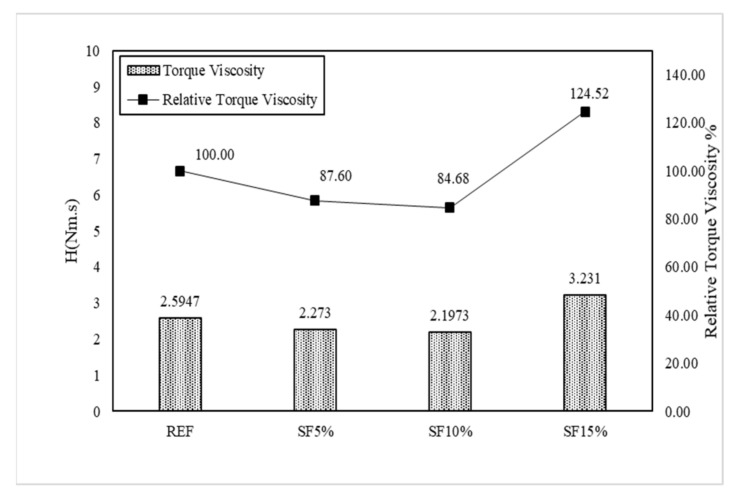
Torque viscosity (H) of the mixing ratio (Jeon’s data).

**Figure 18 materials-15-00428-f018:**
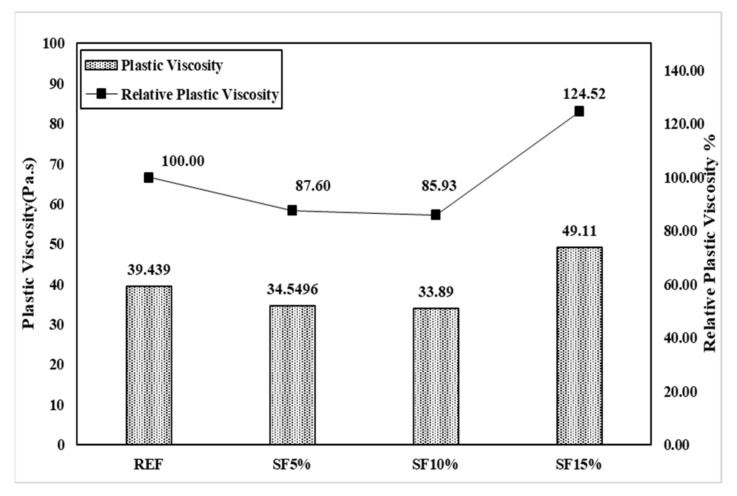
Plastic viscosity (µ) of the mixing ratio (Jeon’s data).

**Figure 19 materials-15-00428-f019:**
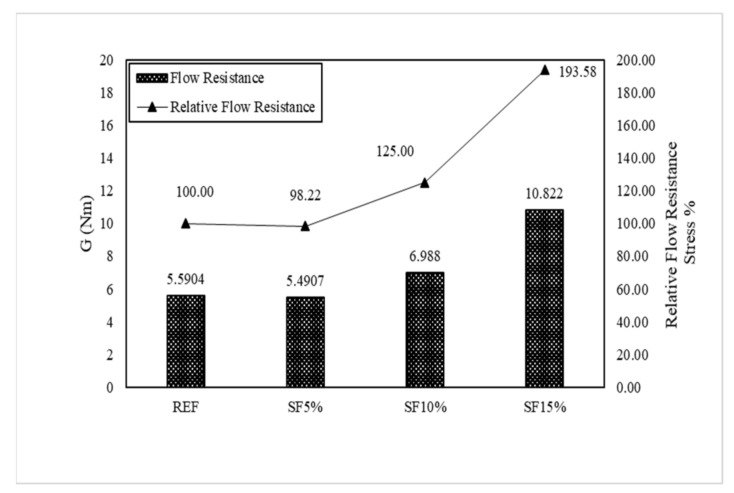
Flow resistance (G) of the mixing ratio (Jeon’s data).

**Figure 20 materials-15-00428-f020:**
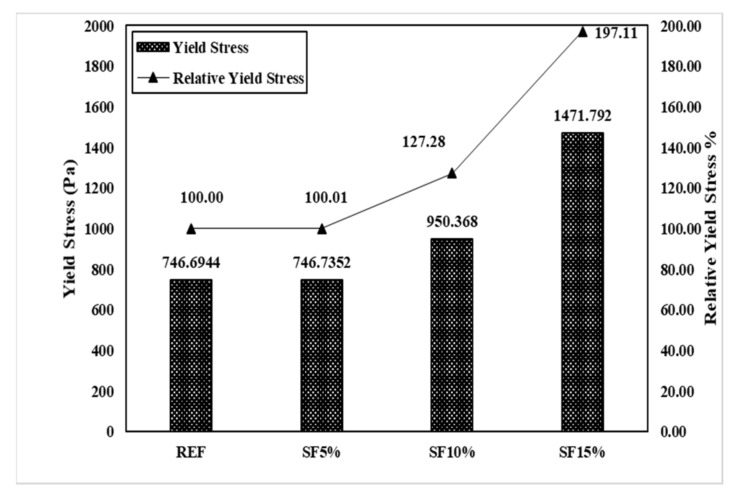
Yield stress (τ_0_) of the mixing ratio (Jeon’s data).

**Figure 21 materials-15-00428-f021:**
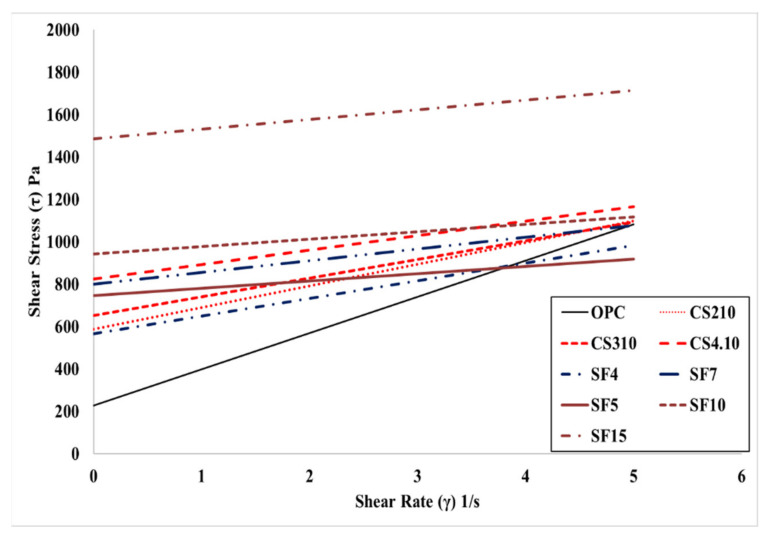
Bingham model result comparison of all mixing ratios (own results and Jeon’s indicated with SF5, SF10, and SF15).

**Figure 22 materials-15-00428-f022:**
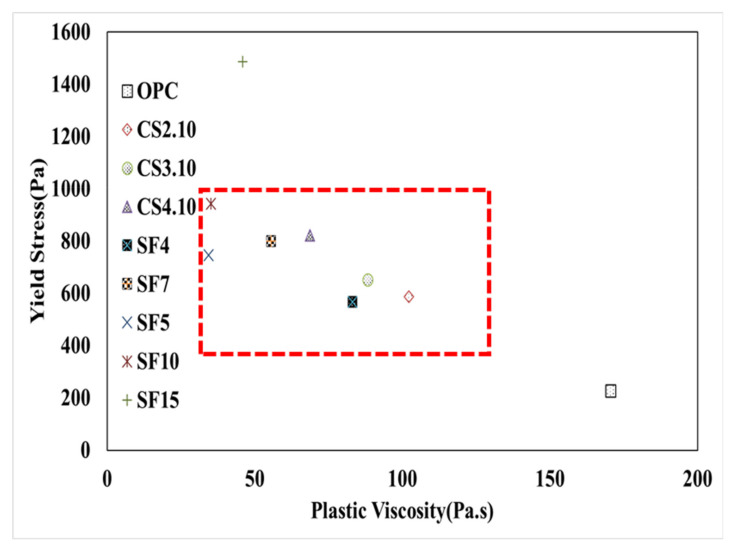
Pumpability identification by the Bingham model result of all mixing ratios (own results and Jeon’s indicated with SF5, SF10, and SF15).

**Figure 23 materials-15-00428-f023:**
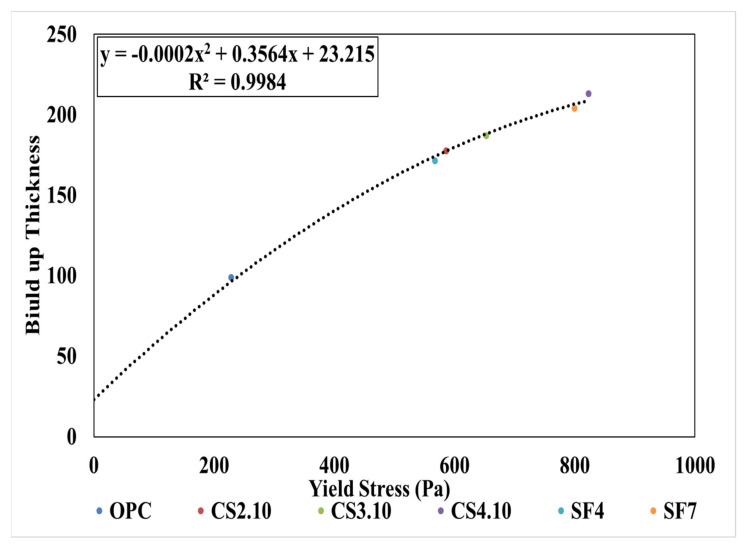
Build-up thickness measuring using Bingham yield stress.

**Table 1 materials-15-00428-t001:** Physical properties of AEWR.

Density (g/cm^3^)	Alkali Amount (kg/m^3^)	Water Reduction Rate (%)	Ratio of Bleeding Amount (%)	Usage
1.05	0.01	22	48	Binder weight × 0.5∼2.5%

**Table 2 materials-15-00428-t002:** Shotcrete mixtures.

Mix Designation	Unit Weight (kg/m^3^)					AE.WR	CS
	W	C	S	G	SF		
OPC	184	460	1218	417		4.14(0.9%)	
CS2.10	179	460	1222	418		6.90(1.5%)	9.2
CS3.10	176	460	1224	419		7.80(1.7%)	13.8
CS4.10	173	460	1227	420		8.28(1.8%)	18.4
SF4	184	442	1213	415	18	5.52(1.2%)	
SF7	184	428	1210	414	32	5.52(1.2%)	
CS: 10: Particle Size 10 nm.							
**CS**				**SF**			
2: Colloidal Silica 2%				4: Silica Fume 4%			
3: Colloidal Silica 3%				7: Silica Fume 7%			
4: Colloidal Silica 4%							

**Table 3 materials-15-00428-t003:** Air volume, slump, and rheology characteristics according to the colloidal silica and silica fume mixing ratio.

Mixtures	Slump (mm)	Air Contents (%)		H(Nm.s)	G(Nm)	τ_0_(Pa)	µ(Pa.s)
		Before Sprayeing	After Sprayeing				
OPC	140	7.2	3.2	8.7	1.45	228.2	170.6
CS2.10	140	7	2.7	6.65	2.87	586.3	102.2
CS3.10	130	7.8	3.4	6.15	3.06	653.1	88.2
CS4.10	125	8.2	3.2	5.5	3.61	823.1	68.7
SF4	140	6	2.6	5.54	2.75	567.2	83.2
SF7	140	6.5	3.1	4.56	3.48	800.1	55.5

**Table 4 materials-15-00428-t004:** Rheology and Bingham model characteristics according to the silica fume mixing ratio (Jeon’s data).

Mixtures	H(Nm.s)	G(Nm)	τ_0_ (Pa)	µ(Pa.s)
RFE	2.59	5.59	746.69	39.44
SF5	2.27	5.49	746.74	34.55
SF10	2.19	6.98	950.37	33.89
SF15	3.23	10.82	1471.79	49.11

**Table 5 materials-15-00428-t005:** Build-up thickness based on yield stress.

Mix Criteria	Yield Stress (Pa)	Build-up Thickness (mm)
OPC	228.2	99.0
CS2.10	586.3	177.6
CS3.10	653.1	187.1
CS4.10	823.1	213.1
SF4	567.2	171.5
SF7	800.1	204.1

## Data Availability

All data, models, and code generated or used during the study are included in the submitted article.

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
