# Peer review of "Rheological Behavior of High-Performance Shotcrete Mixtures Containing Colloidal Silica and Silica Fume Using the Bingham Model"

_materials, 2022, doi:10.3390/ma15020428_

Round 1

Reviewer 1 Report

In the manuscript' Rheological behavior of high-performance shotcrete mixtures containing colloidal silica compared using the Bingham model', the authors present an experimental study on different admixtures for shotcrete, investigating workability in terms of rheological characteristic

The aim of the paper is clear;

However for certain fundamental issues I cannot recommend the paper for publication in the present form:

  • Language is an issue. For instance, already the first sentence of the abstract contains a typo/error: 'Wet-mix shotcrete is a more applicable for construction technology in modern era.'
  • The preparation of the manuscript is poor. Many figures need considerable improvement (Figure 1 is completely malformed), and I cannot read the text in some of , e.g, Figure 8. In general, the figures make the impression that they are taken from another source due to the loss of quality.
  • Basically no  state of the art / literature review on experimental studies on shotcrete and/or admixtures (or for classical concrete) is presented; If there is not state of the art, the authors should mention this explicitly.
  • Only single experimental values are presented; there is no information about statistics, experimental scatter, or the number of experiments. Hence, validation of the results is not ensured.

Please improve these fundamental issues.

Author Response

Reviewer Reply #1:

Thank you for thoroughly reviewing this article and giving the precious comments. I have reviewed your recommendations, improvements and edited the paper based on your comments. After correcting your all comments, I hope to make the work suitable for publication in the Materials-MDPI Journal. Thanks for allowing me to revise this paper and attached below are the all points.

  • Language is an issue. For instance, already the first sentence of the abstract contains a typo/error: 'Wet-mix shotcrete is a more applicable for construction technology in modern era.'

Ans: As per your comments, I am trying to improve the whole paper language and sentence patterns. Here, I also attached the manuscript for your better understanding of improving the language of that manuscript.

  • The preparation of the manuscript is poor. Many figures need considerable improvement (Figure 1 is completely malformed), and I cannot read the text in some of , e.g, Figure 8. In general, the figures make the impression that they are taken from another source due to the loss of quality.

Ans: I changed the Figure 1 because it’s literally malformed. Moreover, I changed the Figure 8 formation for better improvement of the journal paper.

  • Basically no  state of the art / literature review on experimental studies on shotcrete and/or admixtures (or for classical concrete) is presented; If there is not state of the art, the authors should mention this explicitly.

Ans: As per your guidance, I added some extra references for improving the manuscript acceptance.

  1. Only single experimental values are presented; there is no information about statistics, experimental scatter, or the number of experiments. Hence, validation of the results is not ensured.

Ans: Here, we tried to examine the Rheological results so that we are just concerned about these properties without statistical analysis.

Reviewer 2 Report

Dear Authors,

your paper is dealing with an interesting topic.

However, the paper is difficult to read, as you are not familiar with shotcrete terminology, and you should check the English with a native speaker. These problems in language lead also to a paper with lack in structure and many unnecessary repetitions. The paper must be much more straight forward.

I marked the text, which is very unclear in this sense, with yellow.

Some topics you should revise with special care:

  • Do not judge the dry-mix shotcrete methods as you have no experience with it.
  • Make clear in the beginning, that your investigations include no pumping or spraying, only base mix properties! If you made pumping or spraying tests in any way, describe the testing procedure.
  • When adjusting the admixture dosage to reach 120 mm slump, you also influence the rheological parameters. What is the impact of that to your conclusions?
  • A slump of 120 mm is too small for today’s tunnel applications. But in most cases accelerators are used for sufficient fast setting and early strength. You do not specify in which field your shotcrete should be used. In your abstract and introduction you are not clear on that and mix up different applications.
  • What is the purpose of equation (3)? It appears quite isolated and is not explained.
  • The ICAR Rheometer gives you relative values, which may be transferred to absolute values. You use this transfer but be aware of its problems. For most of your diagrams the transfer is not changing the general trend.
  • Instead of using another classic rheometer (IBB), another type of rheometer for judging pumpability should have been used, e.g. SLIPER from Schleibinger testing systems.

Be aware that the rheology of pumping concrete is the rheology of the lubrication layer in the pipe. The build up thickness is evaluated by the rheology you used.

Author Response

Reviewer Reply #2:

Thank you for thoroughly reviewing this article and giving the precious comments. I have reviewed your recommendations, improvements and edited the paper based on your comments. After correcting your all comments, I hope to make the work suitable for publication in the Materials-MDPI Journal. Thanks for allowing me to revise this paper and attached below are the all points.

  1. As per your comments we are trying to improve the language patterns. Moreover, attached is the ediatge certificate for English editing.
  • Make clear in the beginning, that your investigations include no pumping or spraying, only base mix properties! If you made pumping or spraying tests in any way, describe the testing procedure.

Ans: Actually, we used shotcrete for practical use with shotcrete equipment. It means we used pumping and spraying. For that proof, I attached one picture, but we did not describe all processes because wet-mix shotcrete process is very common.

  • When adjusting the admixture dosage to reach 120 mm slump, you also influence the rheological parameters. What is the impact of that to your conclusions?

Ans: Actually, slump 120mm shows the empirical value for workability.  So, rheological properties is needed for showing the actual value for shotcrete performance (workability and shootability) . So, initially showed the empirical properties fixed by the slump 120±30 mm.

  • A slump of 120 mm is too small for today’s tunnel applications. But in most cases accelerators are used for sufficient fast setting and early strength. You do not specify in which field your shotcrete should be used. In your abstract and introduction you are not clear on that and mix up different applications.

Ans: In this study mainly focus on normal structure and wall type of structure not on the tunnel and overhead structure. So, we did not use the accelerators and tried to use different types of mix proportion incorporated with silica fume and colloidal silica for shotcrete performance. According to the Korean standard, for tunnel shotcrete slump must be above 140mm. However, in this study, for normal structure considered slump 120mm.

  • What is the purpose of equation (3)? It appears quite isolated and is not explained.

Ans: The purpose of equation 3 is to measure τ0 and µ in the Bingham Model. This equation is called the Reiner-Riwlin equation. This equation is a modified equation for considering the dead zone in the coaxial cylinders rheometer.

  • The ICAR Rheometer gives you relative values, which may be transferred to absolute values. You use this transfer but be aware of its problems. For most of your diagrams the transfer is not changing the general trend.

Ans: According to your comment, the graph behavior must be the same but the problem in the rheology graph from rheometer data does not intersect Y-axis and it’s not properly linear. Also, for drawing the linear regression line some points are not matched in the line (R2 has differed from mixture to mixture). On the other hand, Bingham model data intersect the Y-axis for measuring the yield stress and plastic viscosity. These data are important for measuring the rheological behavior of concrete. Moreover, Bingham model line is linear and R2 is 1, also in this model the line intersects the Y-axis. So that, we can find yield stress, on the other hand, ICAR and IBB rheometer data did not insect Y-axis. We can estimate the importance of that.

  • Instead of using another classic rheometer (IBB), another type of rheometer for judging pumpability should have been used, e.g. SLIPER from Schleibinger testing systems.

Ans: In here, mainly used Bingham model data for representing pumability and shootability. Moreover, comparison with ICAR and IBB rheometer data. Because of, conversation from IBB value to Bingham model follow: τo= 136 G and µ=15.2 H

and ICAR follows the coaxial cylindrical Reiner-Riwlin equation for computing τo, µ.

  1. IBB and ICAR are the equipment but their calculation for converting Bingham model totally different. We have already discussed previously why the Bingham model is important for interpreting data.
  2. Table 2 already changed as per your comment.
  3. Actually, when transferring the ICAR and IBB data to Bingham data some graphs trends will not be similar but here all graphs trends are similar and more accurate because of totally linear line and connect the Y-axis.
  4. I changed Figure 20 as per your comment.
  5. Here, the pumping rate is described in the equipment specification. Moreover, when we did the shooting, we controlled the pressure by ourselves.
  6. For build-up thickness we used our data and for example, just use Beaupre reference. For build-up thickness, we used for normal structure wall, not overhead structure. Figure 22 shows the relationship between Build up thickness and yield stress. Because increasing the yield stress means adhesiveness also increases. So, Yield stress increased build-up thickness also increased simultaneously.

Round 2

Reviewer 1 Report

The authors addressed some of the mentioned issues;

However, English language and figures still require additional effort.

I did not explicitly list of all of the figures which require additional preparation, since it must be clear to the authors that e.g., also Figure 6 (left) is simply not readable?

Regarding:

>>> Ans: Here, we tried to examine the Rheological results so that we are just concerned about these properties without statistical analysis. <<<

This is not a question of the focus of the paper; The authors present  single measurement results (1 result per composition) without any discussion on the uncertainty.

Author Response

Reviewer Reply #1:

Thank you for thoroughly reviewing this article and giving the precious comments. I have reviewed your recommendations, improvements and edited the paper based on your comments. After correcting your all comments, I hope to make the work suitable for publication in the Materials-MDPI Journal. Thanks for allowing me to revise this paper and attached below are the all points.

As per your comments, I am trying to update the sentence pattern and all figures for making visible.

Reviewer 2 Report

Dear Authors,

your revised paper has some minor improvements, but overall it is still lacking structure, sufficient English and critical discussion. It cannot be published in this form in an international  magazine.

The paper is difficult to read and you should check the English with a native speaker. In the text I highlighted the worst problems only. These problems in language lead also to a paper with lack in structure and many unnecessary repetitions. The paper must be much more straight forward.

Some topics you should revise with special care:

  • Make clear in the beginning, that your investigations include no documentation on pumping or spraying, only base mix properties! If you made pumping or spraying tests in any way, describe the testing procedure in more detail and add the picture.
  • Make clear in Tables and Figures who is the author!
  • Specify the Silica Fume and Colloidal Silica you used in your tests
  • When adjusting the admixture dosage to reach 120 mm slump, you also influence the rheological parameters. What is the impact of that to your conclusions? Report the dosage used for each test.
  • When discussing possible build-up of layer thickness you have to address the influence of your 7% of air after spraying. You investigate a mix with 7% air, sprayed you have 3% of air.
  • What is the purpose of equation (3)? It is not explained. E.g. what stands Ω for? I know, that you want to show Reiner-Riwlin
  • The ICAR Rheometer gives you relative values, which may be transferred to absolute values. You use this transfer but are aware of its problems. Did you use commercial software? For most of your diagrams the transfer is not changing the general trend.
  • Instead of using another classic rheometer (IBB), another type of rheometer for judging pumpability should have been used, e.g. SLIPER from Schleibinger testing systems.
  • With ICAR you can only asses the possible layer thickness of a mix which will not change air content during spraying, not the pumping behaviour. You mention this fact now in the last sentences of your paper (lubrication layer).

Author Response

Reviewer Reply #2:

Thank you for thoroughly reviewing this article and giving the precious comments. I have reviewed your recommendations, improvements and edited the paper based on your comments. After correcting your all comments, I hope to make the work suitable for publication in the Materials-MDPI Journal. Thanks for allowing me to revise this paper and attached below are the all points.

  1. Did you perform that? You do not report any testing details. Rather you compared with data from literature.

Ans: In this study, we performed shooting but we did not perform any special test for pumpabilty checking. Actually, pumpability and shootability can measure by the rheology data. So, this is our main foucus to do that.

  1. You did it or compared with data from literature? In your comments you answer, that you pumped and sprayed. But this is not visible in this Paper.

Ans:  We added some pictures in the main manuscript and describe little bit about that.

  1. 1) Wording

2) Bingham is a simpe rheological model. ICAR is a rheometer. You can apply Bingham on ICAR results too. You maybe want to adress relative results from ICAR. Explain better

Ans: Yes, ICAR is a rheometer and Bingham is a simple rheological model. Moreover, to convert data from ICAR or IBB to Bingham model differently due to value conversion of flow resistance to yield stress and torque viscosity to plastic viscosity. For that reason, we consider that Bingham model is more appropriate rather than any kind of rheometer data.

  1. Quality control is not depending on nozzle size.

Ans: Quality of the shotcrete depends on the nozzle sizes. Because of, if the nozzle diameter plays a vital role of shooting and quality of the concrete.

  1. List is OK, but what did you learn from their investigation for your tests.

Ans : Investigation added in the main manuscript.

  1. IBB and ICAr are the equipment. Bingahm is the simplest model to interpret data.

Ans: Already answered I think.

  1. But you do not explain in detail how you get the absolut values from the relative values, nor do you adress the difficulties of this transformation. There is also literature, that says, that this transformation is not possible. Did you use comercial software from company  German?

Ans: In here, we used the solver which is developed by the ICAR manufacturing company. But, anybody can develop that solver in excel by modifying the Reiner-Riwlin equation by data interpolation.

  1. But for this purpose you used a rheometer, which will not give you the proper results. See the last sentence of your paper.

Ans: In this paper, you tried to mention that conversation to ICAR data to Bingham model data. Data conversion depends on the coaxial cylinder and dead zone considering by using solver. On the other hand, you mentioned that ICAR R2 results are more acceptable in the world. It’s true but all the points are in the line so that we did the linear regression. Moreover, in the Bingham Model all data is fully linear. Only for that reason, we mention that comments.

  1. In our test we used silica fume differnet 4 and 7% respectively and colloidal silica 2,3 and 4% respectively (10nm).
  2. Seems to be calculated with 70 Liter/m³ of air. Which is reasonable, but not after spraying. Be aware of that for your conclusions and lyare thickness build up

Ans :  We used the percentage of AEWR for the proper mix design of the shotcrete by trail. And, we did the shooting for our test.

  1. Commercail software by German? You did not test the reliability

Ans: We used the solver which is provided by the company with ICAR rheometer.

  1. These two graphs show the same trend. But the transformation from relative values to the absolut values is tricky and may be defective.

Ans: ICAR and Bingham graphs show the same trend because of ICAR data get from the T=GH+N that equation. After that, ICAR data was interpreted by the company-provided solver. After that getting data τo and µ. Than simply use the Bingham model equation to draw that. For example: when γ= 0 that time τ= τo. Then, put γ=1,2,3 or any number than we can get τo with fixed plastic viscosity (µ).

  1. In figure 11 to 14, compare data with OPC and draw in the same figure. So, it’s connected with CS4,10 and SF4. However, comparing is the main issue here.
  2. Jeon's result, not your own. Yes, these are Jeon’s data and I already mentioned in here.
  3. Figure 21 is our data and Jeon’s data.
  4. How do you know, which parameter settings are necessary for good pumping?

You did not pump or spray it. in your comment, you write, that you pumped and sprayed. But this is not visible in this paper.

Ans: For analysis of the pumpability, using Yield stress vs Plastic viscosity graph. This data was collected from the Bingham model. After that, compared with practical work we can find out which data are pumpable and not pumpable. Beaupre and Tattershall used that and draw the workability box.

  1. yield stress from own tests, Build-up thickness calculated according to Beaupre ???

Ans: This data from our practical work.

  1. You mentioned before that you tested air content after spraying but you do not report on the results

Ans: I added this data in table 3.

  1. Conclusion little bit edited based on your comments.
  2. Did you spray with 4.6 m³/h or did you take this from literature?

Ans: We did it.

  1. When adjusting the admixture dosage to reach 120 mm slump, you also influence the rheological parameters. What is the impact of that to your conclusions?

Ans: Actually, slump 120mm shows the empirical value for workability.  So, rheological properties are needed for showing the actual value for shotcrete performance (workability and shootability) . So, initially showed the empirical properties fixed by the slump 120±30 mm.

  1. A slump of 120 mm is too small for today’s tunnel applications. But in most cases accelerators are used for sufficient fast setting and early strength. You do not specify in which field your shotcrete should be used. In your abstract and introduction you are not clear on that and mix up different applications.

Ans: In this study mainly focus on normal structure and wall type of structure not on the tunnel and overhead structure. So, we did not use the accelerators and tried to use different types of mix proportion incorporated with silica fume and colloidal silica for shotcrete performance. According to the Korean standard, for tunnel shotcrete slump must be above 140mm. However, in this study, for normal structure considered slump 120mm.

  1. Make clear in Tables and Figures who is the author!

Ans: I think it’s all cleared in that paper.

  1. Specify the Silica Fume and Colloidal Silica you used in your tests

Ans: It’s already mentioned how many percentage SF and colloidal are used in this study.

  1. With ICAR you can only asses the possible layer thickness of a mix that will not change air content during spraying, not the pumping behaviour. You mention this fact now in the last sentences of your paper (lubrication layer)

Ans: It’s just a future plan for extra research. Moreover, ICAR rheometer gave flow resistance and torque viscosity data which are used for measuring the pumpability and shootability. Because ICAR data was used to measure yield stress and plastic viscosity. I already cleared in here.

Round 3

Reviewer 2 Report

Dear Authors,
your third revision is a little better and some aspects are clearer now. 
But you should check the English once more. I could not correct everything. 
Find my comments in the text.

Author Response

Thank you for thoroughly reviewing this article and giving the precious comments. I have reviewed your recommendations, improvements and edited the paper based on your comments. After correcting your all comments, I hope to make the work suitable for publication in the Materials-MDPI Journal. Thanks for allowing me to revise this paper and attached below are the all points.

 In here, R2 is 0.984 is realistic but all points are not in the line so that we are trying to convert this data in Bingham model and trying to say that it's quite correct.

I think after that, a reviewer will satisfy.

Also, we are trying to correct reviewer’s correcting as per comments in the main manuscript.
